# α-Gal Nanoparticles Mediated Homing of Endogenous Stem Cells for Repair and Regeneration of External and Internal Injuries by Localized Complement Activation and Macrophage Recruitment

**DOI:** 10.3390/ijms231911490

**Published:** 2022-09-29

**Authors:** Uri Galili, Josef W. Goldufsky, Gary L. Schaer

**Affiliations:** Division of Cardiology, Department of Medicine, Rush University Medical College, Chicago, IL 60612, USA

**Keywords:** alpha-gal nanoparticles, natural anti-Gal antibody, wound healing, myocardium regeneration, macrophage recruitment, stem-cell homing

## Abstract

This review discusses a novel experimental approach for the regeneration of original tissue structure by recruitment of endogenous stem-cells to injured sites following administration of α-gal nanoparticles, which harness the natural anti-Gal antibody. Anti-Gal is produced in large amounts in all humans, and it binds the multiple α-gal epitopes (Galα1-3Galβ1-4GlcNAc-R) presented on α-gal nanoparticles. In situ binding of anti-Gal to α-gal nanoparticles activates the complement system and generates complement cleavage chemotactic-peptides that rapidly recruit macrophages. Macrophages reaching anti-Gal coated α-gal nanoparticles bind them via Fc/Fc receptor interaction and polarize into M2 pro-reparative macrophages. These macrophages secrete various cytokines that orchestrate regeneration of the injured tissue, including VEGF inducing neo-vascularization and cytokines directing homing of stem-cells to injury sites. Homing of stem-cells is also directed by interaction of complement cleavage peptides with their corresponding receptors on the stem-cells. Application of α-gal nanoparticles to skin wounds of anti-Gal producing mice results in decrease in healing time by half. Furthermore, α-gal nanoparticles treated wounds restore the normal structure of the injured skin without fibrosis or scar formation. Similarly, in a mouse model of occlusion/reperfusion myocardial-infarction, near complete regeneration after intramyocardial injection of α-gal nanoparticles was demonstrated, whereas hearts injected with saline display ~20% fibrosis and scar formation of the left ventricular wall. It is suggested that recruitment of stem-cells following anti-Gal/α-gal nanoparticles interaction in injured tissues may result in induction of localized regeneration facilitated by conducive microenvironments generated by pro-reparative macrophage secretions and “cues” provided by the extracellular matrix in the injury site.

## 1. Introduction

Amphibians such as salamander, newt, and axolotl and fish such as zebra fish display the amazing capacity of spontaneous regeneration of amputated limbs, injured myocardium, and other tissues. The regeneration process includes the formation of a blastema which is a mass of cells capable of growth and regeneration into organs or body parts. The blastema contains cells of mesenchymal origin which migrate to the injury site, proliferate, and dedifferentiate into stem cells that restore the morphology and function of the amputated limb or injured tissue [1]. The homing of the blastema cells into the injury site is preceded by extensive migration of macrophages to the injury site [1,2,3,4,5,6]. These macrophages secrete a variety of cytokines which are thought to orchestrate the process of blastema formation in tissue, limb or organ regeneration [1,2,7] and in injured heart regeneration [3,4,5,6,8]. The exact mechanisms mediating the regeneration in zebra fish and amphibians are not fully clear as yet and seems to involve complex cross-talks between various cells. Similar regeneration is not found in adult mammals, with very few exceptions, like injured liver [9] and finger-tip in mice [10]. In adult mammals, most other external injuries such as skin wounds [11] and internal injuries like myocardial injury following myocardial infarction (MI) also involve infiltration of macrophages into the injured tissue [3,12]. However, the healing process is completed by the default mechanism of fibrosis and scar formation, rather than by regeneration of the injured tissues. Fibrosis and scar formation in skin wounds can isolate the inner tissues from pathogens in the surrounding environment. In healing of ischemic myocardium after infarction, fibrous tissue and scar formation prevent rupture of the left ventricular (LV) wall. However, this may result in adverse remodeling and dilation of the LV, leading to congestive heart failure and death in patients post-MI. 

In contrast to fibrosis and scar formation in post-MI adult mice, neonatal mice (1–2 day old) [13,14,15] and neonatal pigs [16,17] display full regeneration of injured myocardium, whereas myocardial injury 7 days or more after birth, results in fibrosis and scar formation similar to that observed in adult mice and humans [13,14,15,16,17]. Regeneration of the injured heart in neonatal mice was found to be preceded by extensive migration of macrophages into the injury site [15]. The mechanism responsible for myocardial regeneration in neonates has not been fully elucidated. Nevertheless, these observations led to the suggestion that in order to achieve regeneration in adult mammals, one must “turn back the cardiac regenerative clock”, i.e., find methods which resurrect the neonatal regenerative potential in adults [18]. The extensive migration of macrophages into the injured myocardium in both neonatal and adult mice, resulting in regeneration vs. scar formation, respectively, suggests that there are qualitative differences between the macrophages migrating into injured tissues in neonates compared to adults [19]. This review describes a method we have developed for “turning back the regenerative clock” in adult mice, i.e., inducing migration of pro-reparative macrophages into injured tissues. These macrophages orchestrate regeneration of injured tissues instead of scar formation. 

Clues that helped in identifying requirements for inducing regeneration in adult mice have been the determination of processes that are found to be common in the spontaneous regeneration observed in zebra fish, amphibians (i.e., salamander, newt or axolotl) and neonatal mice. Two such common processes are: (i) Extensive recruitment of macrophages to the injury site, as described above [4,7,8,14]; (ii) Localized activation of the complement system in the injury site. While the complement system is usually considered to be part of the protective immune response against various pathogens, it plays an important role also in regenerative processes. Studies in zebra fish, amphibians, and neonatal mice with various injuries, including heart injuries, have demonstrated activation of the complement cascade, upregulation of complement receptors, expression of C5a receptor-1 (C5aR1) and of C3aR1 in cells at the injury site and prevention of regenerative processes in the absence of the complement cleavage peptides C5a and C3a or of their corresponding receptors [20,21,22,23,24,25]. In view of the upregulation of complement activities in injuries and the concomitant extensive migration of macrophages to injury sites in fish, amphibian, and neonatal mice, our objective has been to develop a method for localized activation of the complement system and recruitment of macrophages in injuries in adult mice. We further aimed to determine whether such induced processes can result in regeneration of skin wound without scar formation and in post-MI regeneration of ischemic ventricular wall in mice. We achieved the objectives of localized activation of the complement system and recruitment of macrophages by the use of nanoparticles called “α-gal nanoparticles” which bind one of the most abundant natural antibodies in humans, called the “anti-Gal” antibody. In anti-Gal producing mice, administration of α-gal nanoparticles to skin wounds and into post-MI ischemic myocardium was found to induce complement activation, macrophage and stem cell recruitment and regeneration of injured skin and myocardium without significant scar formation.

## 2. Anti-Gal and the α-Gal Nanoparticles

Anti-Gal is one of the most abundant natural antibodies in humans, constituting ~1% of immunoglobulins [26,27,28,29,30]. It is produced throughout life, starting few months after birth [26,31], in response to antigenic stimulation by gastrointestinal bacteria [32,33,34,35,36,37]. Anti-Gal binds specifically to a carbohydrate antigen called the “α-gal epitope” with the structure Galα1-3Galβ1-4GlcNAc-R [29,38,39,40,41]. Whereas anti-Gal is naturally produced in all humans, apes, and Old-World monkeys (monkeys of Asia and Africa), the α-gal epitope is synthesized as ~10^5^–10^7^ epitopes/cell in nonprimate mammals (marsupials and placentals), lemurs (living in Madagascar) and New-World monkeys (monkeys of South-America), all of which lack the anti-Gal antibody [42,43,44]. The α-gal epitope is synthesized in the Golgi apparatus on nascent carbohydrate chains (glycans) of glycoproteins and glycolipids by the glycosylation enzyme α1,3galactosyltransferase (α1,3GT). The α*1,3GT* gene, called *GGTA1*, was inactivated in ancestral Old-World primates 20–30 million years ago resulting in lack of α1,3GT activity in Old-World monkeys, apes and humans and production of the natural anti-Gal antibody [43,45]. Binding of anti-Gal to α-gal epitopes results in strong activation of the complement system, as indicated by the effective cytolytic activity of serum anti-Gal on cells presenting multiple α-gal epitopes [46,47,48].

The production of the natural anti-Gal antibody in all humans provides a unique opportunity for harnessing the immunological potential of this antibody for development of novel therapies in various clinical settings. These therapies are called α-gal therapies because they use the α-gal epitope in different contexts in order to manipulate the anti-Gal antibody [49]. One of the tools developed for this purpose is α-gal nanoparticles [49,50,51,52,53,54,55]. These nanoparticles are biodegradable, sub-microscopic liposomes constructed from phospholipids, cholesterol and glycolipids of which α-gal glycolipids are the majority. Their production is rather simple since they are presented in large numbers on rabbit red blood cell (RBC) membranes. There are ~2 × 10^6^ α-gal epitopes per each rabbit RBC [56]. Cell membranes are obtained from rabbit RBC lysed in water. These RBC membranes are kept overnight in a stirred solution containing chloroform and methanol for extraction of the phospholipids, cholesterol and glycolipids, while the proteins are precipitated and removed by filtration. The extract is dried and resuspended in saline by sonication to form liposomes. Additional sonication by a sonication probe breaks these liposomes into nanoparticles that present ~10^14^ α-gal epitopes per mg nanoparticles (Figure 1A) [51]. When incubated in human serum, anti-Gal IgM, IgG and IgA antibodies readily bind to these nanoparticles, and activate the complement cascade similar to many other antigen/antibody interactions [51]. α-gal nanoparticles may be prepared also from synthetic α-gal glycolipids and other α-gal glycoconjugates combined with phospholipids or other biodegradable materials used for nanoparticles formation.

## 3. In Situ Effects of α-Gal Nanoparticles

### 3.1. Hypothesis

Anti-Gal binds very effectively to the α-gal nanoparticles because of the high concentration of α-gal epitopes on these nanoparticles. This binding can be readily detected even at human serum dilutions of >1:100. The antibody binding further activates the complement cascade [48] that produces the complement cleavage peptides C5a and C3a which are among the most potent chemotactic factors for macrophage recruitment in the body [51]. Therefore, we hypothesized that application of α-gal nanoparticles to wounds, or injection of these nanoparticles into myocardium injured by ischemia, results in binding of anti-Gal to α-gal epitopes on the nanoparticles (Step 1 in Figure 1B). Anti-Gal is released with other blood components from ruptured capillaries in the wound or from capillaries ruptured by the syringe needle. This anti-Gal/α-gal nanoparticles interaction activates the complement cascade in the area surrounding the α-gal nanoparticles, causing formation of C5a and C3a chemotactic peptides that induce migration of macrophages into that area (Step 2 in Figure 1B). Once the macrophages reach the α-gal nanoparticles they bind via their Fc receptors (FcR) the Fc portion of anti-Gal coating the nanoparticles (Step 3 in Figure 1B). This interaction activates the macrophages and stimulates them to secrete pro-reparative cytokines and growth factors which together with the high local complement activation induce local repair of the injury and regeneration of the injured tissue, similar to that observed in neonatal mice (Step 4 in Figure 1B). Based on studies demonstrating that homing of stem cells to injured sites is mediated by the interaction between C5a and C3a and their corresponding receptors on the stem cells [58,59,60,61], it was further hypothesized that these complement cleavage peptides generated in Step 1 direct stem cell homing to the site of the α-gal nanoparticles. In addition, it was hypothesized that among the multiple cytokines secreted by the activated macrophages in Step 4 [51] there may be a cytokine(s) which contributes to the homing of stem cells to the treated injured area. Upon reaching to the injury site, the stem cells may proliferate and receive “cues” from the microenvironment and adjacent uninjured cells to differentiate into cells that regenerate the tissue into its original structure and function.

### 3.2. Experimental Animal Models

The in vivo effects of α-gal nanoparticles cannot be studied in the usual experimental animal models such as mouse, rat, guinea pig or rabbit, because these species, as well as other non-primate mammals synthesize the α-gal epitope as self-antigen, and do not produce the anti-Gal antibody. This problem was solved following the generation of mice [62,63] and pigs [64,65] in which the *GGTA1* gene encoding α1,3GT was disrupted (i.e., knocked out). The resulting α1,3GT knockout (GT-KO) mice lack α-gal epitopes and can produce anti-Gal in titers comparable to those in humans following immunization with xenogeneic cell membranes presenting multiple α-gal epitopes. A convenient source for such immunizing cell membranes was found to be pig kidney membrane (PKM) homogenate [66]. The mice do not produce sufficient anti-Gal without immunization, probably because of the sterile conditions of their housing and sterile food they receive. In contrast GT-KO pigs, which live in nonsterile conditions develop a gastrointestinal flora that induces natural production of anti-Gal by the age of 6 weeks and later [48,67,68]. Most studies discussed in this review were performed in GT-KO mice that received 3–5 immunizations with PKM (~50 mg per injection) in order to induce anti-Gal production at titers comparable to those in humans.

### 3.3. Recruitment of Macrophages by α-Gal Nanoparticles

Injury of tissues usually results in recruitment into the wound of macrophages by chemotactic factors, such as macrophage inflammatory protein-1 (MIP-1), monocyte chemoattractant protein-1 (MCP-1), and regulated on activation, normal T cell expressed and secreted (RANTES), released from damage cells within and around the wound and from endothelial cells at the injury sites [69,70,71,72,73]. In order to determine whether α-gal nanoparticles binding anti-Gal generate complement chemotactic peptides capable of inducing effective recruitment of macrophages, these nanoparticles were injected into various healthy tissues and the histology of the injected sites was evaluated at various time points. As seen in Figure 2, intradermal injection of the nanoparticles in anti-Gal producing GT-KO mice resulted in extensive recruitment of macrophages within 24 h (Figure 2A). Addition of cobra venom factor (a potent inhibitor of complement activation) to the α-gal nanoparticles completely inhibited this recruitment of the macrophages [51]. The great majority of the recruited cells were macrophages as confirmed by their positive staining with anti-F4/80 antibody, performed 4 days post injection (Figure 2B). This antibody stains specifically macrophages in mice. The macrophages kept increasing in numbers till Day 7 (Figure 2C). At that time point, the macrophages were large with ample cytoplasm, and their cell membranes were close to each other. Morphology of individual macrophages observed at the injection site, 7 days post-injection is shown in Figure 2D. The macrophages displayed a size of ~15–30 μm with multiple vacuoles, most of which could be formed as a result of extensive uptake of the anti-Gal coated α-gal nanoparticles. The recruited macrophages maintained this morphology and large numbers even by Day 14. However, after 21 days no macrophages were found at the injection site and the skin displayed normal histologic structure [51].

Recruitment of macrophages was also observed 4 days post two injections of α-gal nanoparticles into healthy myocardium of an anti-Gal producing GT-KO mouse (arrows in Figure 2E) and after an injection near a branch of the sciatic nerve in the mouse thigh (Figure 2F). Injection of saline resulted in no recruitment of macrophages in any of these tissues (not shown). These findings indicated that administration of α-gal nanoparticles into various tissues in the body is likely to induce local robust activation of the complement cascade due to anti-Gal/α-gal epitopes interaction and results in effective recruitment of macrophages by these nanoparticles.

### 3.4. Characterization of the Recruited Macrophages as M2 Polarized Macrophages

Macrophages recruited by α-gal nanoparticles were further studied with biologically inert sponge discs (made of polyvinyl alcohol- PVA 10 mm diameter, 3 mm thickness). These sponges were loaded with ~10 mg of α-gal nanoparticles in saline and implanted subcutaneously in anti-Gal producing GT-KO mice. The sponge discs were explanted after 7 days, the cells within them retrieved by repeated squeezing in saline, washed, counted, and subjected to flow cytometry analysis. Approximately 0.5 × 10^6^ cells were retrieved from each sponge disc, all presenting a morphology as that of the macrophages in Figure 2D, whereas control sponge discs containing only saline had <10% infiltrating cells (~0.2 × 10^5^ cells/disc) [50]. Flow cytometry analysis of the cells recruited by α-gal nanoparticles further confirmed that most of them (>90%) displayed the macrophage markers CD11b and CD14, but no markers of T or B cells (Figure 3A) [50]. These macrophages were further evaluated for M1 (pro-inflammatory) or M2 (pro-reparative) polarization. As shown in Figure 3B, most macrophages displayed the distinct M2 markers IL10 and Arginase-1 and they lacked any IL12 staining. The latter marker is found in M1 polarized macrophages. It is of note that the largest macrophages displayed a higher intensity of IL10 and Arginase-1 staining than smaller sized macrophages, suggesting increased production of these markers as the macrophages become larger and increase their activation state.

The extensive uptake of α-gal nanoparticles suggested in Figure 2D was assumed to be the result of opsonization due to the binding of the Fc “tail” of anti-Gal coating the α-gal nanoparticles to Fc receptors (FcR) on the macrophages as illustrated in Step 3 of the hypothesis in Figure 1B. Indeed, incubation of macrophages for 2 h with anti-Gal coated α-gal nanoparticles resulted in firm binding of the nanoparticles to macrophages (Figure 4A), whereas α-gal nanoparticles not coated with anti-Gal display no binding to the macrophages (Figure 4B). Subsequent incubation of the macrophages binding anti-Gal coated α-gal nanoparticles for 24 h and 48 h indicated that the ensuing activation of these cells stimulates them to secrete vascular endothelial growth factor (VEGF) which induces neo-vascularization in injured tissues. No significant secretion above background level was observed in cultures of macrophages that did not bind α-gal nanoparticles in the absence of anti-Gal coating the nanoparticles [51,52].

### 3.5. Homing of Stem Cells

As indicated above, there is an extensive migration of macrophages into injury sites prior to regeneration of amputated limbs in amphibians, or in injured hearts in zebra fish, amphibians, and neonate mice [1,2,3,4,5,6,7,8,9,10,11,12,13,14,15]. In addition, homing of stem cells is associated with interaction of the complement cleavage peptides C5a and C3a [58,59,60,61]. These two sets of findings raised the question of whether, in addition to the macrophage recruitment by α-gal nanoparticles into the PVA sponge discs, can the anti-Gal/α-gal nanoparticles interaction direct homing of stem cells into the sponge discs. For this purpose, cell suspensions obtained on Day 7 from implanted PVA discs containing α-gal nanoparticles (Figure 2D), were cultured as 10^5^ cells/mL and subsequently inspected for colonies formation, which may suggest presence of stem cells. The macrophages, which were the large majority of cells in these cultures displayed no proliferating ability. However, after 5 days of culturing, large colonies of proliferating cells were observed at a frequency of one colony per 50,000 to 100,000 cultured cells (Figure 5). These findings raised the possibility that the macrophages recruited by the α-gal nanoparticles into the PVA sponge discs and/or the complement cleavage peptides induced homing of stem cells capable of forming colonies. The colonies contained 300–1000 cells per colony, suggesting that the cells forming the colonies were stem cells which proliferated at a cell-cycle time of ~12 h.

The identity of the proliferating cells forming colonies as stem cells, is further supported by the co-expression of three stem cell markers Sca1, CD29 and CD105 on most of these cells analyzed by flow cytometry (unpublished observations). Overall, the combined complement activation, recruitment, and polarization of M2 macrophages by the anti-Gal/α-gal nanoparticles interaction seems to be followed by recruitment of a small number of stem cells to the site of the nanoparticles. These recruited stem cells display an extensive ability to proliferate. α-gal nanoparticles treatment was further found to induce regeneration in several in vivo regeneration induction models, as described below.

### 3.6. Cartilage Regeneration in PVA Sponge Discs

In preliminary studies, PVA sponge discs were loaded with a suspension containing 10 mg α-gal nanoparticles and microscopic fragments of porcine meniscus cartilage, kept frozen for several years [57]. The fragmented cartilage was devoid of live cells and was pre-treated with α-galactosidase in order to eliminate α-gal epitopes [74]; thus, to enable binding of anti-Gal only to α-gal epitopes on the nanoparticles. Control sponge discs contained only meniscus cartilage fragments devoid of α-gal epitopes. It was assumed that homing of stem cells into the sponge disc, induced by the α-gal nanoparticles, will be followed by proliferation of these cells. The vicinity or accidental contact of these stem cells with the cartilage extracellular matrix (ECM) may direct them to differentiate into fibroblasts secreting collagen fibers and other ECM components. The sponge discs containing α-gal nanoparticles and fragmented cartilages, or only fragmented cartilage, were implanted subcutaneously for 5 weeks. Subsequently, the sponge discs were explanted, sectioned, and stained by hematoxylin and eosin (H&E) and by Masson trichrome in which the formed collagen fibers in the ECM are stained blue.

The round sponge disc is shown in Figure 6A with two areas of chondrogenesis, marked by rectangles. The lower rectangle is magnified in Figure 6B demonstrating the dark nuclei of fibrochondroblasts interspaced within the large areas of the fibrocartilage ECM. The large amounts of de novo synthesized collagen fibers are further demonstrated in their dark blue staining by Masson trichrome and the interspaced relatively few nuclei of fibrochondroblasts stained purple in Figure 6C,D. In these figures, the PVA sponge walls of the spaces in which chondrogenesis occurred are blue gray and the space between these walls and the cartilage tissue is an artifact due to shrinking of the cartilage tissue that is dehydrated during the staining process of the sections. The complementary shape of sponge cavities and that of the edge of the tissue indicates that the growth of the tissue was limited by these walls. The features of the actual meniscus cartilage are shown in Figure 6F, in which the fibrochondroblasts are organized in parallel orientation and the fibrocartilage ECM organized in a similar orientation among the cells producing it. Such an organization is not feasible within the sponge discs because of the highly irregular shape of the PVA walls bordering the cavities in the sponge. Sponge discs containing cartilage homogenate, but no α-gal nanoparticles displayed no chondrogenesis and the cavities contained mostly fat tissue (Figure 6E). Overall, these preliminary studies suggested that α-gal nanoparticles administered together with an ECM homogenate may recruit stem cells, resulting in differentiation of the proliferating stem cells into the original tissue producing the tested ECM [57].

## 4. Regeneration of Skin Wounds by α-Gal Nanoparticles Prevents Scar Formation

The physiologic mechanism for healing of wounds in the skin includes migration of pro-inflammatory macrophages into the injured tissue which debride the wound of dead cells and ECM. This stage is followed by the appearance of pro-reparative macrophages that orchestrate the migration of fibroblasts into the wound. These fibroblasts generate a dense fibrotic tissue, resulting in formation of scar tissue that isolates the inner tissues from pathogens on the skin [11,75,76]. Thus, it was of interest to determine whether application of α-gal nanoparticles to skin wounds in anti-Gal producing GT-KO mice has any effect on wound healing and regeneration of the injured skin.

Full thickness wounds in oval shape with dimensions of ~6 × 9 mm were formed in the mice under anesthesia (Figure 7Aa). A suspension of 10 mg/mL α-gal nanoparticles was applied to the wound by the use of a spot bandage covered with that suspension (referred to as “treated wounds”). Control wounds were treated with saline covered spot bandage. The wounds were inspected at various days post treatment and further subjected to histological analysis. The extent of wound healing was determined as proportion (%) of the wound area that was covered with regenerating epidermis.

Treated wounds displayed after 3 days ~40% covering of the wound surface with regenerating epidermis, whereas control wound displayed no significant epidermal growth (Figure 7B). Histology of the treated wounds further demonstrated extensive infiltration of macrophages into the wound bed of treated wounds whereas only very few macrophages were observed in control wounds [51,52]. By Day 6, the treated wounds displayed complete or near complete healing indicated by 95–100% covering of the wound surface with the regenerating epidermis, whereas control wounds displayed only ~20% growth of the epidermis. Even after 12 days, the extent of epidermis regeneration in control wounds was only 60% and full healing of control wounds was usually observed after 14–16 days. These observations implied that treatment of wounds with α-gal nanoparticles decreased the wound healing time by at least 50%. It is of note that α-gal liposomes (i.e., nanoparticles with larger size, prior to probe sonication) also displayed an accelerating effect on wound healing, but at a slower pace, possibly because the small and numerous nanoparticles better dispersed into all remote areas of the wound bed [51]. Treatment of the wounds with nanoparticles prepared from RBC membranes of GT-KO pigs (i.e., nanoparticles lacking α-gal epitopes) resulted in a healing pace that was not significantly different from that observed in control mice treated with saline (Figure 7B).

Accelerated healing by α-gal nanoparticles was also observed in independent studies on wound healing in GT-KO mice [53], in thermal injuries [50] and in radiation skin injuries [55]. Accelerated healing of skin injuries was also observed in GT-KO pigs treated with these nanoparticles [77]. In addition, chronic wounds in diabetic mice healed within 12 days following application of α-gal nanoparticles to these wounds, whereas no healing was observed with saline covered spot bandages [52,54].

Treated and control wounds were further examined 28 days post wounding in order to determine whether the treatment had any effect on the final structure of the healed wound. At that time point, all wounds were completely healed. Control wounds displayed the typical histology of fibrosis and scar formation of healed wound, consisting of dense connective tissue, as indicated by the deep blue staining of high-density collagen in Masson trichrome staining (Figure 8). In addition, no skin appendages growth (e.g., hair and sebaceous glands) was observed, and the epidermis displayed hyperplasia, as indicated by multiple layers of epidermal cells. In contrast, the treated wounds displayed clear regeneration of the normal skin structure which includes thin epidermis, loose connective tissue in the dermis, as well as regrowth of skin appendages such as hair shafts, sebaceous glands, fat tissue and smooth muscle in the hypodermis (Figure 8).

The clear differences between the scar formation in the saline treated wounds and regeneration of normal skin structure without scar formation in α-gal nanoparticles treated wounds, strongly suggest that the accelerated healing of these wounds also involves the recruitment of skin or mesenchymal stem cells by the localized activation of the complement cascade and the rapid recruitment of macrophages that are activated within the treated wounds. In addition to secretion of cytokines that accelerate the regeneration of the epidermis for covering the wound, these activated macrophages secrete VEGF that induce rapid vascularization of the wound [51,52,78]. The recruited macrophages were further found to display activation of fibroblast growth factor (FGF), interleukin 1 (IL1), platelet derived growth factor (PDGF), and colony stimulating factor (CSF) genes [51]. These cytokines, together with other pro-reparative cytokines secreted from these macrophages and with complement cleavage peptides C5a and C3a, may induce homing of stem cells which reach the wound via the newly developed vascular system and generate a microenvironment conducive to regeneration of the injured skin. The absence of scar tissue in the treated wounds also suggests that the regenerative processes activated by the α-gal nanoparticles treatment are initiated prior to the activation of the default fibrosis processes, thereby avoiding the mechanisms involved in scar formation. It remains to be determined whether this conducive microenvironment is more effective in enabling the endogenous recruited stem cells to proliferate and differentiate into skin cells than direct exogenous application of stem cells to wounds [78]. In addition, it may be possible that the continuous contribution of the activated macrophages in production of localized pro-reparative cytokines/growth factors may result in more effective healing and regeneration processes than exogenous cytokines administered to wounds [79]. The exogenous cytokines may diffuse from the wound and have a shorter half-life than the wide range of cytokines continuously secreted by the activated macrophages recruited into the wound by α-gal nanoparticles.

## 5. Regeneration of Myocardium by α-Gal Nanoparticles Post-Myocardial Infarction

The accelerated healing of skin wounds, the regeneration of normal skin structure and prevention of scar formation described above, raised the question of whether a similar treatment with α-gal nanoparticles may prevent scar formation and restore normal myocardial structure and function after myocardial infarction (MI). MI is the leading cause of death in the USA and is caused by the sudden occlusion of a coronary artery and subsequent death of cardiomyocytes. Myocardial necrosis post-MI heals by the default repair mechanism of fibrosis and scar formation. Although this healing process reduces the risk of death from spontaneous rupture of the LV wall, the absence of any significant myocardial regeneration results in reduced contractility, adverse ventricular remodeling and LV dilation which can lead to congestive heart failure and premature death.

In order to determine whether α-gal nanoparticles treatment can prevent post-MI scar formation and restore the normal structure and function of the myocardium, the mid-left anterior descending (LAD) coronary artery in mice was occluded for 30 min, followed by reperfusion. The treated mice then received two 10 μl injections of α-gal nanoparticles (10 mg/mL) into the injured myocardium, 2 mm from each side of the reperfused LAD, and control mice received two 10 μl saline injections. Inspection of the hearts in control mice after 28 days indicated that this 30 min ischemia resulted in distinct fibrosis and scar formation in the left ventricular myocardium and thinning of the ventricular wall (Figure 9) [19]. In contrast, the mice treated with α-gal nanoparticles injections demonstrated a near complete regeneration of the ventricular wall and the injured myocardial territory was repopulated with cardiomyocytes displaying normal structure and function (representative hearts in Figure 9A) [19]. Echocardiographic assessment of myocardial contractility 7 days post-MI demonstrated a marked reduction in the left ventricle contractility and dilation of the LV chamber dimension in both control and treated groups. Echocardiographic re-assessment after 28 days demonstrated continued poor contractility in the control group, whereas mice treated with α-gal nanoparticles displayed restoration of normal contractility and normal LV chamber dimension similar to that measured prior to the coronary infarction [19]. Planimetry measurements indicated that on average, the damage by fibrosis and scar formation on Day 28 was 10 folds higher in post-MI saline treated hearts than in hearts treated with α-gal nanoparticles (Figure 9B).

It is of interest to note that macrophage infiltration was observed in both post-MI hearts treated with saline and with α-gal nanoparticles. In the control hearts treated with saline, the peak of macrophage infiltration was observed on Day 4 (Figure 10E); whereas in α-gal nanoparticles treated hearts the peak infiltration was on Day 7 (Figure 10B) and few mitotic figures were observed (magnification of red circles in Figure 10C). Staining by bromo-deoxy-uracil (BrDU) or by identification of proliferating cells nuclear antigen (PCNA) displayed insufficient number of stained cells for concluding that there was extensive cell proliferation within the regenerating myocardium on Days 9 and 11 post-MI [19]. The macrophages in the two groups seem to differ from each other, as implied from the ventricle morphology in Day 14. In the saline treated hearts, the ventricular wall was thinner than normal and contained fibrotic tissue (Figure 10F); whereas in α-gal nanoparticles treated hearts, most of the normal structure of the ventricular wall was regenerated and fibrosis was sparse. (Figure 10D).

Overall, these observations strongly suggest that α-gal nanoparticles treatment recruits and activates macrophages that “turn back the regenerative clock” in adult mice, by preventing scar formation, as in skin wounds, and by orchestrating regeneration of ischemic myocardium into normal structure and function. In contrast, macrophages recruited by the injured myocardium in saline treated hearts mediate the default process of fibrosis and scar formation, similar to their effects in saline treated skin wounds. These observations further support the assumption that the post-MI myocardium regeneration in adult mice may be associated with α-gal nanoparticles induced homing of stem cells, but they do not directly prove this assumption. The number of resident or mesenchymal stem cells in the heart is very small, therefore their evaluation by immune staining is difficult. A direct proof for the possible regeneration of post-MI myocardium by mesenchymal stem cells may require performance of stem cell lineage tracing using biologically labeled stem cells (e.g., green fluorescent protein labeled stem cells) administered into anti-Gal producing GT-KO mice that underwent ischemia/reperfusion and treated with α-gal nanoparticles following the reperfusion.

## 6. Concluding Remarks

Migration of many macrophages to injury sites and activation of the complement system are common phenomena observed in the regenerative processes of fish, amphibians, and neonatal mice. These common features led to the assumption that regenerative processes may be induced also in adult mouse injuries, resulting in restoration of the normal structure and function of the original tissue, rather than the occurrence of the default process of fibrosis and scar formation. Thus, it was of interest to determine whether localized complement activation and subsequent recruitment and activation of macrophages in injury sites could affect stem cell homing and result in regeneration of injured tissues in adult mice. Such localized activation of the complement system and recruitment and activation of macrophages became feasible by the use of α-gal nanoparticles interacting with the natural anti-Gal antibody, which is an abundant antibody in all humans. The α-gal nanoparticles present multiple α-gal epitopes which are the ligand of anti-Gal. Thus, administration of α-gal nanoparticles to any tissue or to skin wounds results in binding of anti-Gal to the nanoparticles and activation of the complement system as in most antigen/antibody interactions. The resulting formation of chemotactic complement cleavage peptides C5a and C3a is followed by extensive recruitment and activation of macrophages which bind the α-gal nanoparticles via Fc/Fc receptors interaction. In addition, various studies have demonstrated homing of stem cells due to interaction between C5a and C3a with their corresponding receptors on these cells. Following their activation, the recruited macrophages secrete a variety of pro-reparative cytokines, including VEGF and stem cell recruiting factors. The recruited stem cells display high proliferating activity with cell cycle time of only ~12 h. When α-gal nanoparticles are applied to wounds in anti-Gal producing mice, they initiate processes that result in decreased healing time by half and in regeneration of the skin structure, including reappearance of skin appendages, thereby preventing fibrosis and scar formation processes which are observed in untreated wounds. Similarly, injection of α-gal nanoparticles into post-MI ventricular wall, injured by ischemia, results in near complete regeneration of the injured myocardium, whereas hearts injected with saline display fibrosis and scar formation. The suggested homing of stem cells under the effects of α-gal nanoparticles may be applicable also to injuries in various tissues in which accelerated regeneration could prevent fibrosis and scar formation, thereby restoring the original structure and function of the injured tissue.

## Figures and Tables

**Figure 1 ijms-23-11490-f001:**
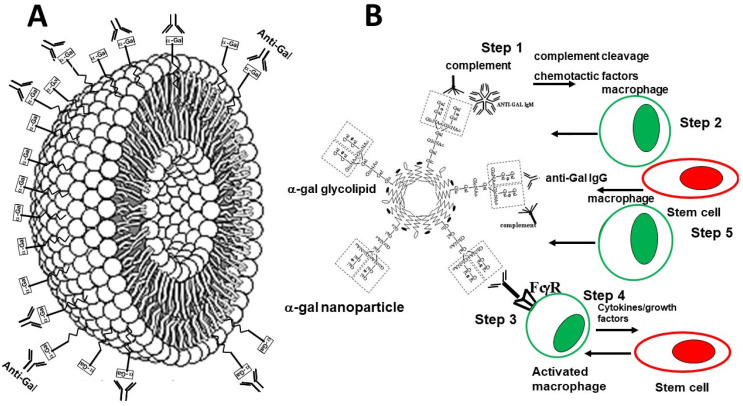
Illustration of the α-gal nanoparticle as a spheric lipid bilayer studded with α-gal glycolipids (α-gal epitopes illustrated as rectangles on glycolipid molecules) (**A**) and of the hypothesized outcomes of anti-Gal binding to α-gal epitopes on the nanoparticles (**B**). The various steps, following anti-Gal/α-gal nanoparticles interaction, are detailed below. Macrophages are colored green and stem cells-red. Modified from ref. [57].

**Figure 2 ijms-23-11490-f002:**
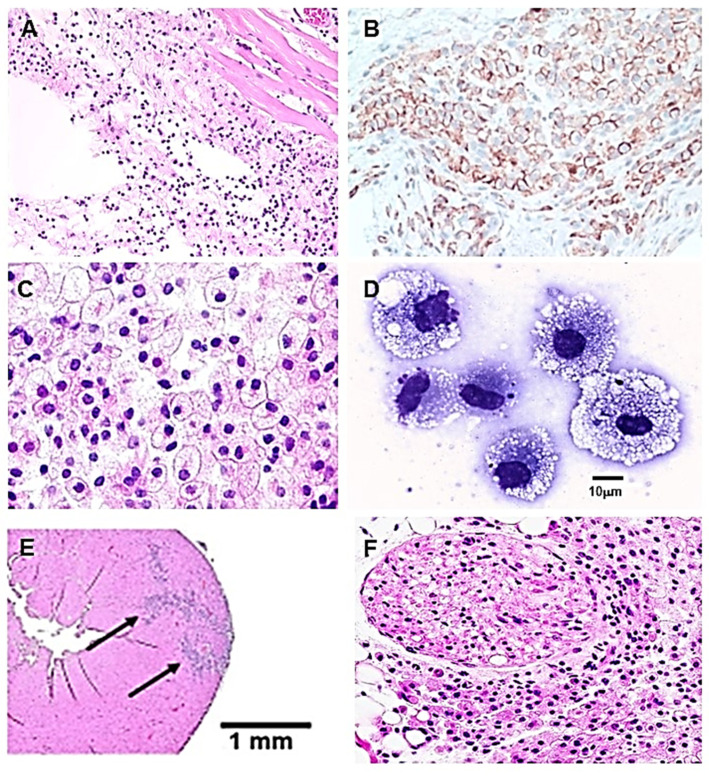
Demonstration of macrophage recruitment by α-gal nanoparticles, in various tissues of anti-Gal producing α1,3galactosyltransferase knockout (GT-KO) mice. (**A**) Macrophage recruitment 24 h after intradermal injection of 10 mg α-gal nanoparticles. The empty oval area is the space formed by the injection of α-gal nanoparticles. The nanoparticles were dissolved by alcohol during the processing for hematoxylin & eosin staining (H&E × 100). (**B**) Macrophages identified at the injection site, 4 days post injection, by specific staining with the anti-F4/80 antibody coupled to peroxidase (HRP) (×200). (**C**) The intradermal injection site after 7 days is full of many large macrophages containing vacuoles, suggesting activation of these cells (H&E × 400). (**D**) Individual macrophages, similar to those in (**C**) migrating into polyvinyl alcohol (PVA) sponge disc containing 10 mg α-gal nanoparticles and implanted subcutaneously into GT-KO mouse for 7 days. The many vacuoles in the cytoplasm of the macrophages are of the anti-Gal coated α-gal nanoparticles internalized by the macrophages (Wright staining, ×1000). (**E**). Mouse healthy heart, 4 days following two injections of α-gal nanoparticles (each 10 μl of 10 mg/mL nanoparticles). Recruited macrophages are identified by arrows (H&E × 20). (**F**). Infiltration of macrophages into an area near a sciatic nerve branch (oval structure), 4 days post injection of α-gal nanoparticles (H&E × 200). Modified from ref. [57].

**Figure 3 ijms-23-11490-f003:**
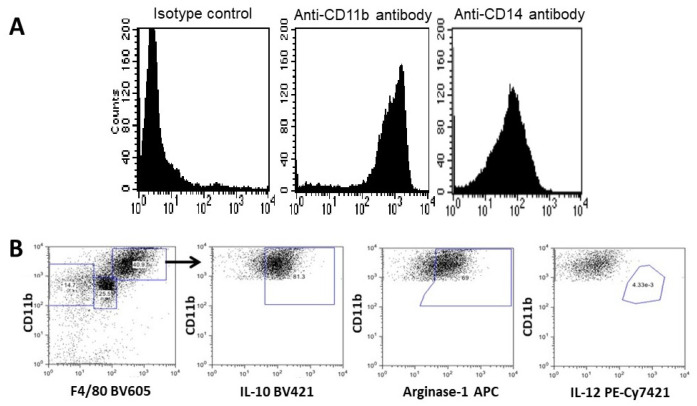
Flow cytometry of cells recruited into PVA sponge discs containing α-gal nanoparticles. (**A**). The large majority of the cells is macrophages as indicated by the staining of most cells with antibodies to two biomarkers of macrophages, CD11b and CD14. No staining was observed for T or B lymphocytes (not shown). (**B**). Analysis of the recruited macrophages state of polarization. The large size macrophages (CD11b^pos^/F4/80^pos^) were positive also for IL-10 and Arginase-1 but were negative for IL-12, implying that the majority of the recruited cells were M2 macrophages. Modified from ref. [57].

**Figure 4 ijms-23-11490-f004:**
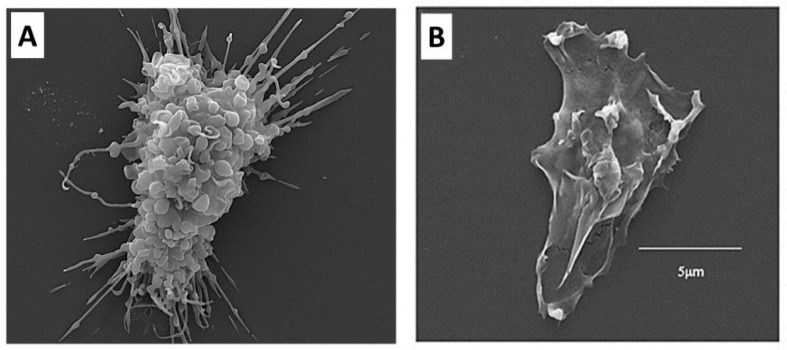
Scanning electron microscopy (SEM) of macrophages. (**A**). An adherent macrophage lacking α-gal epitopes and incubated with α-gal nanoparticles coated with the anti-Gal antibody, then washed and processed for SEM. Note the multiple α-gal nanoparticles binding to the macrophage via Fc/Fc receptor interactions and covering the surface of the macrophage. (**B**). A macrophage incubated with α-gal nanoparticles in the absence of anti-Gal. No nanoparticles bind to the macrophage cell membrane in the absence of the antibody. Adapted with permission from ref. [57].

**Figure 5 ijms-23-11490-f005:**
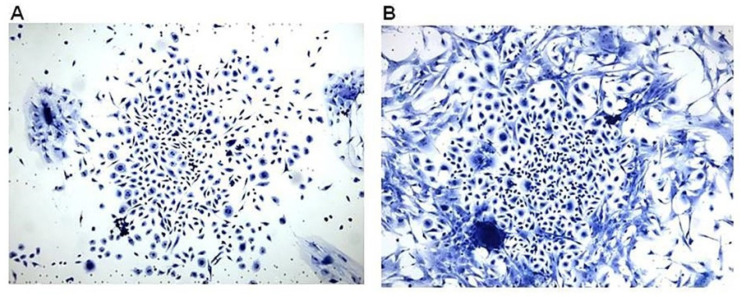
Representative examples of colonies formed within 5 days of culturing cells obtained from PVA sponge discs containing 10 mg/mL α-gal nanoparticles that were implanted subcutaneously for 7 days in GT-KO mice producing anti-Gal. The frequency of colony forming cells was approximately one in 5 × 10^4^ and 1 × 10^5^ recruited macrophages. The number of cells per colony was ~300 (**A**) and ~1000 (**B**) (×100). Reproduced with permission from ref. [57].

**Figure 6 ijms-23-11490-f006:**
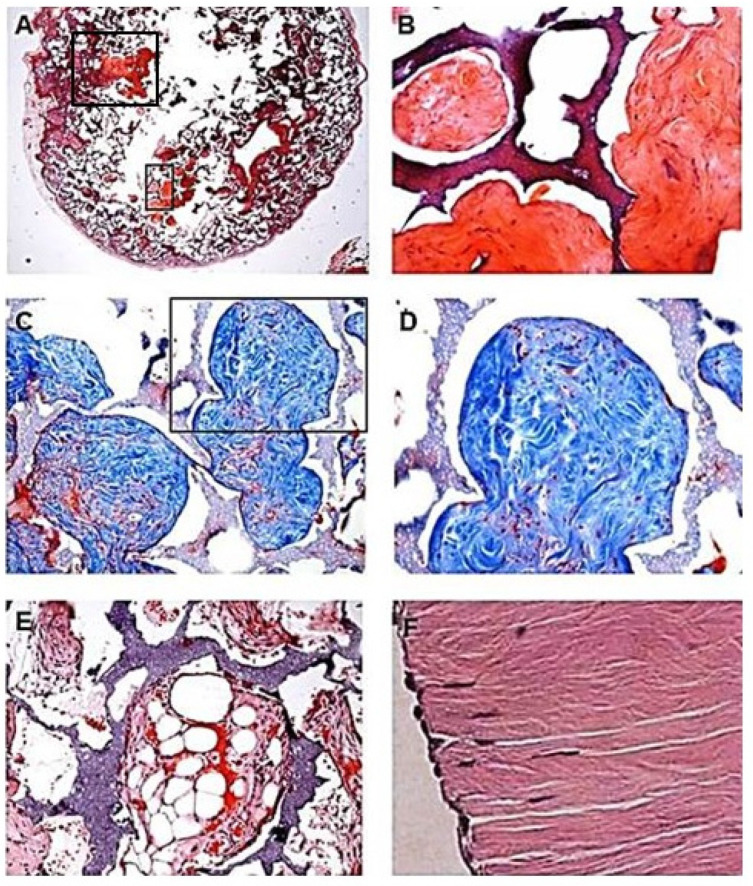
Formation of fibrocartilage in PVA sponge discs containing α-gal nanoparticles (10 mg/mL) and microscopic porcine meniscus cartilage homogenate as ECM (50 mg/mL), implanted subcutaneously for 5 weeks in anti-Gal producing GT-KO mice. (**A**). PVA sponge disc section for demonstration of fibrocartilage growth (stained red) in areas marked with rectangles (H&E ×10). (**B**). Magnification of the inset in (**A**) demonstrating the fibrocartilage growth (red). The PVA sponge material is stained purple-red. (**C**). Mason-trichrome staining of the collagen fibers blue in an area similar to that in (**B**) (×100). (**D**). The inset in (**C**) (×200) demonstrates fibrocartilage formation consisting of multiple collagen fibers. The nuclei of the few fibrochondroblasts are stained purple. (**E**). Control sponge disc containing only meniscus cartilage ECM homogenate, displaying adipocytes and no fibrocartilage formation (Mason-trichrome × 100). (**F**). Structure of porcine meniscus fibrocartilage displaying parallel organization of fibrochondrocytes and the collagen fibers they produce (H&E, ×200). Representative sections are from 5 GT-KO mice per group. Reprinted with permission from ref. [57].

**Figure 7 ijms-23-11490-f007:**
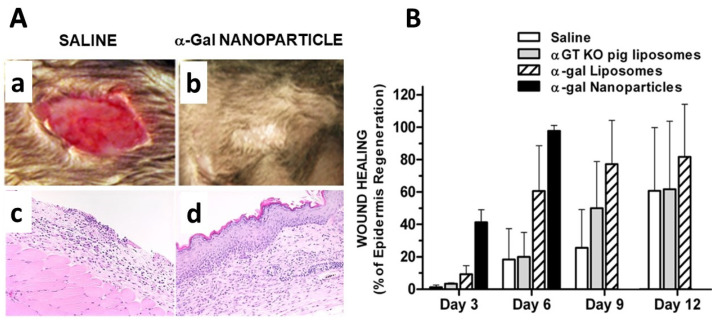
Effect of topical application of α-gal nanoparticles on healing of wounds, determined as proportion of regenerating epidermis covering the wound. (**Aa**) Representative morphology on Day 6 of saline treated oval 6 × 9 mm full thickness excisional wounds. No healing and regeneration are observed. (**Ab**). Wound as in (**Aa**), treated with α-gal nanoparticles. Note the complete healing of the wound which is covered by regenerating epidermis. (**Ac**) Histology of the wound in (**Aa**) (H&E × 100). (**Ad**) Histology of the wound in (**Ab**) (H&E × 100). (**B**) Healing of wounds was evaluated with α-gal nanoparticles (10 mg/mL) (closed columns), α-gal liposomes, which are large α-gal nanoparticles (1–10 μm, hatched columns), 10 mg GT-KO pig liposomes lacking α-gal epitopes (gray columns), or saline (open columns). Mean + SD from ≥5 mice/group. On Day 6, 20 mice were evaluated per each group. Modified from ref. [57].

**Figure 8 ijms-23-11490-f008:**
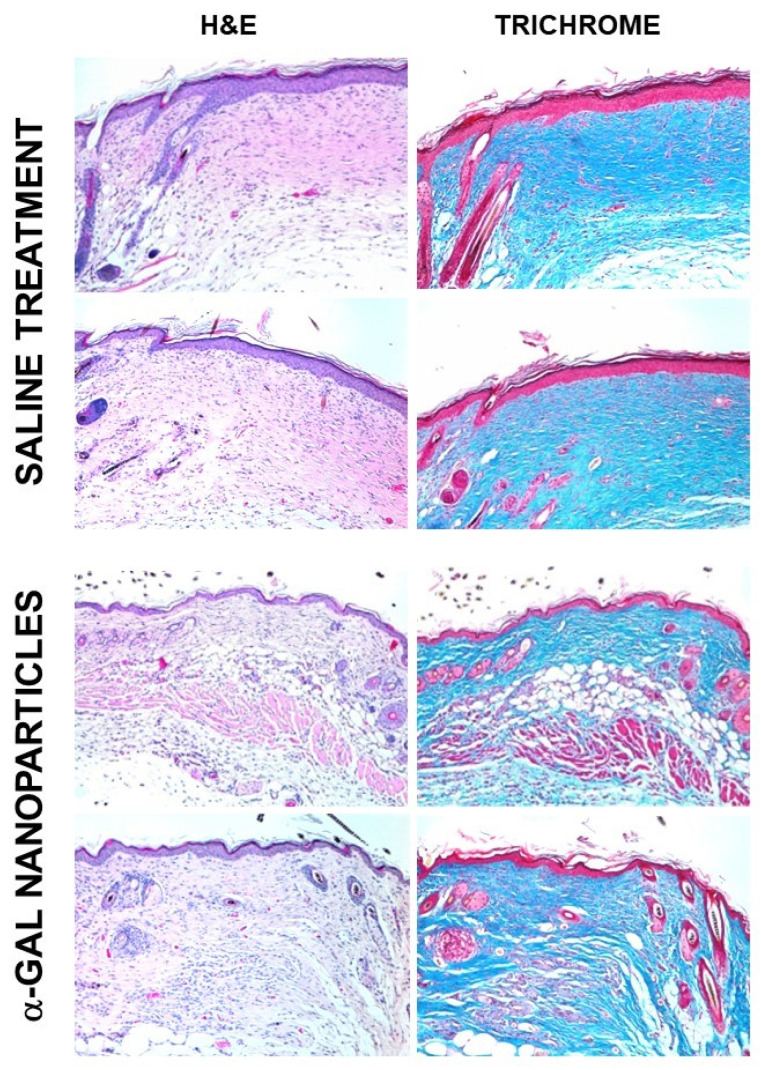
Histology of GT-KO mouse wounds treated with α-gal nanoparticles or with saline and inspected 28 days post treatment. Wounds were stained with H&E or with Masson trichrome for detection of de novo produced collagen. Whereas the saline treated wounds display distinct fibrosis and scar formation (dense connective tissue, hyperplastic epidermis, absence of skin appendages), the α-gal nanoparticles treated wounds display restoration of normal skin structure including growth of hair, loose connective tissue, adipose tissue and smooth muscle cells (×100). Reprinted from ref. [57] with permission.

**Figure 9 ijms-23-11490-f009:**
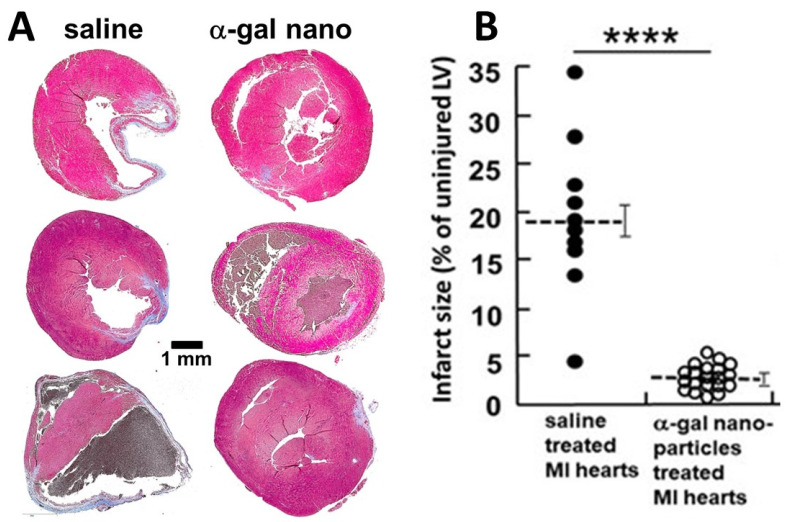
GT-KO mouse hearts undergoing MI by occlusion/reperfusion, treated post-MI with two intramyocardial injections of 10 μl saline or α-gal nanoparticles (10 mg/mL) and inspected 28 days post treatment. (**A**). The hearts were sectioned and stained with Masson trichrome for identification of scar tissue. Representative saline treated hearts displayed scar formation and thinning of the ventricular wall whereas α-gal nanoparticles treated hearts displayed near complete regeneration of the ventricular wall. (**B**). Planimetry studies of fibrosis and scar formation in hearts from 10 saline treated and 20 α-gal nanoparticles treated hearts. Mean ± S.E., **** *p* < 0.0001. Adapted from ref. [19].

**Figure 10 ijms-23-11490-f010:**
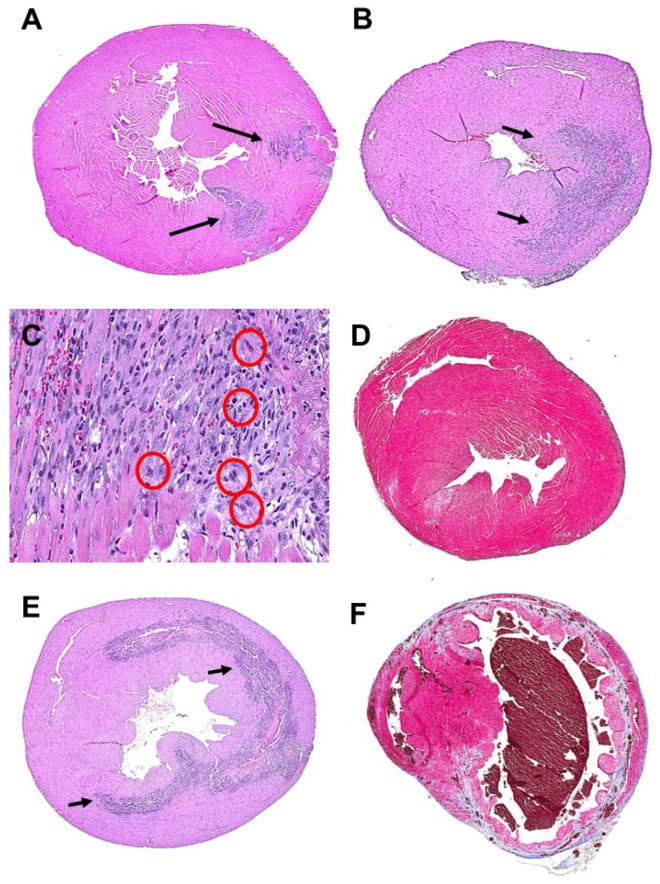
Regeneration (**A**–**D**) or fibrosis (**E**,**F**) of GT-KO mouse hearts undergoing MI by occlusion/reperfusion. (**A**). Heart treated with α-gal nanoparticles, 4 days post treatment. Arrows indicated macrophage infiltration in the two injection sites (H&E × 10). (**B**). Heart treated with α-gal nanoparticles, 7 days post treatment. Arrows indicated the peak infiltration of macrophages (H&E × 10). (**C**). Magnification of infiltration site in heart as in (**B**). Red circles mark mitotic figures at various stages of cell cycle (H&E × 200). (**D**). Near complete regeneration of a treated heart after 14 days (Masson trichrome × 10). (**E**). Saline treated heart after 4 days displays peak infiltration of macrophages, marked by arrows (H&E × 10). (**F**). Saline treated heart after 14 days displaying thinning of the anterior ventricular wall which also displays fibrosis, indicated by the blue-grey stained collagen (Masson trichrome × 10). Modified from ref. [19].

## Data Availability

All the data that support the findings of the study are openly available at https://pubmed.ncbi.nlm.nih.gov/ in manuscripts indicated in this review and detailed in list of References.

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
