# Peer review of "α-Gal Nanoparticles Mediated Homing of Endogenous Stem Cells for Repair and Regeneration of External and Internal Injuries by Localized Complement Activation and Macrophage Recruitment"

_ijms, 2022, doi:10.3390/ijms231911490_

Round 1

Reviewer 1 Report (Previous Reviewer 3)

The authors revised de manuscript in accordance with my comments and suggestions. 

Reviewer 2 Report (Previous Reviewer 1)

Even the previous review did not refer to MSCs, too much parts of this review are overlapped with the previous one and the novelty of this review is really limited. I still cannot recommend this review for publication.

Reviewer 3 Report (Previous Reviewer 4)

Thank you to the authors for considering my comments. Best of luck.

This manuscript is a resubmission of an earlier submission. The following is a list of the peer review reports and author responses from that submission.

Round 1

Reviewer 1 Report

The first author published the similar review recently (https://pubmed.ncbi.nlm.nih.gov/28289553/). The figures and contents are very similar. I admit that several parts are different. However, I cannot recommend this article for publication because of the limited novelty.

Author Response

The manuscript ijms-1880471 has been solicited as a review by the editors of the  Special IJMS Issue "Application of Nanomaterials in Stem Cell Based Therapies" who indicated that  “this manuscript perfectly matches the aim and scope of our special issue” (see attached below email of Ms. Erris Liu of [email protected]). The review mentioned by Reviewer #1 was published in 2017 and does not contain at all the term “stem cell”. Since that publication which describes wound and burn healing by alpha-gal nanoparticles we increased our understanding regarding the effects of these nanoparticles on stem cells. One of the aspects of alpha-gal nanoparticles on stem cells in regenerative processes is the accelerated healing of wounds and the prevention of fibrosis and scar formation. To explain this connection, we have to include in the current review our previous observations on wound healing which were also described in the review mentioned by the reviewer. For this reason, manuscript ijms-1880471 is categorized as a "review" and not as an "original article". To provide a comprehensive picture to the readers on the possible connection between stem cells and injury therapy by alpha-gal nanoparticles, we think that the effects of the nanoparticles on wound healing have to be included in the present review. This association between alpha-gal nanoparticles, stem cells, accelerated wound healing, and prevention of scar formation is novel and was not known in 2017 or in earlier years.

​​​----- Forwarded Message -----

​​​​From: "Erris Liu" <[email protected]>

To: "uri galili" <[email protected]>

Cc: [email protected]

Sent: Tuesday, April 19, 2022 10:17:41 AM

Subject: Re: Paper Invitation: [IJMS] (IF: 5.924, ISSN: 1422-0067) — Special Issue "Application of Nanomaterials in Stem Cell Based Therapies"

Dear Dr. Galili,

Apology for the late reply. Our community is having a Covid-19 situation and it caused some delays. 
After a preliminary check, we are pleased to inform you that this
manuscript perfectly matches the aim and scope of our special issue. We are also glad to hear that you have found funding to support the publication. Since we have applied for a waiver for you, you could keep it for your future papers submitted to our journal.

We look forward to receiving the manuscript. Should you have further queries, please do not hesitate to ask.
Stay safe and healthy.
Kind regards,

Ms. Erris Liu,
Assistant Editor
E-Mail: [email protected]
Skype: live:.live:83525281

Reviewer 2 Report

The work is well done and the topic is essential to consider by scientists and researchers. 

Author Response

We thank the reviewer for the positive view regarding the manuscript.

Reviewer 3 Report

Summary: The manuscript by Galili et all. represents an overview of their own and others’ research findings concerning the tissue regeneration by recruitment of endogenous stem cells to lesion sites after administration of a-gal nanoparticles harnessing the immunological activity of the natural anti-Gal antibody. They also emphasized that in situ binding of anti-Gal to the administered a-gal nanoparticles can activate the complement system and generate chemotactic peptides that recruit macrophages to the nanoparticles site. There has also been shown that these macrophages are polarized into M2 pro-regenerative phenotype and secrete cytokines that orchestrate the healing of the injured tissue,

Following are my comments for authors' consideration:

1. Abstract: The authors are advised to limit the Abstract content to 200 words as recommended in the IJMS Instructions for authors.

2. Introduction: This section appropriately provides the scientific context of the approached topics. However, I would suggest the authors to better emphasize the main aim of the manuscript and, eventually, the foremost conclusion(s) at the end of this section.

3. Please expand the abbreviations at their first appearance within the text even though they are well known. For instance: “FGF, IL1, PDGF, CSF” etc.

4) Typos/spelling errors:secret” (line 45); “expression C5a receptor1” (line 87); “These finding” (line 290)

Author Response

Responses to the specific comments

  1. Abstract: The authors are advised to limit the Abstract content to 200 words as recommended in the IJMS Instructions for authors.

Response: We succeeded in decreasing the number of words in the Abstract from 320 to 250. Because of the various aspects of influence of alpha-gal nanoparticles on stem cells we cannot further decrease to size of the Abstract. We respectfully ask to keep the Abstract at a size of 250 words. Otherwise, the Abstract will become fragmented without distinct continuity.  

  1. Introduction: This section appropriately provides the scientific context of the approached topics. However, I would suggest the authors to better emphasize the main aim of the manuscript and, eventually, the foremost conclusion(s) at the end of this section.

Response: The last part of the Introduction (starting in line 91) was revised according to the suggestions of the reviewer.

  1. Please expand the abbreviations at their first appearance within the text even though they are well known. For instance: “FGF, IL1, PDGF, CSF” etc.

Response: All abbreviations were expanded as requested

4) Typos/spelling errors: “secret” (line 45); “expression C5a receptor1” (line 87); “These finding” (line 290)

Response: We thank the reviewer for pointing out the typos which were corrected.

Reviewer 4 Report

In this review, the authors discuss a novel experimental approach for the regeneration of tissue structure by recruitment of endogenous stem cells to injured sites following administration of a-gal nanoparticles, which harness the immunological activity of the natural anti-Gal antibody and “turn back the regenerative clock” in adult mice.

The manuscript is well written and interesting.

I would suggest only minor cosmetic changes.

Lines 81-83:” Two such common processes are: 1. Extensive […] [4,7,8,14], and 2. […] injury site.” I would number the processes with i) and ii) and separate them with a semi-colon (“Two such common processes are: i) Extensive […] [4,7,8,14], and ii) […] injury site.”) to prevent confusion with references.

Figure 1 would gain in clarity if a color-based distinction could be done between macrophages and stem cells.

Author Response

I would suggest only minor cosmetic changes.

Lines 81-83:” Two such common processes are: 1. Extensive […] [4,7,8,14], and 2. […] injury site.” I would number the processes with i) and ii) and separate them with a semi-colon (“Two such common processes are: i) Extensive […] [4,7,8,14], and ii) […] injury site.”) to prevent confusion with references.

Response: Revised as suggested

Figure 1 would gain in clarity if a color-based distinction could be done between macrophages and stem cells.

Response: Revised as suggested